# Classification of multicategory edible fungi based on the infrared spectra of caps and stalks

**Rui Gao**[1◉], **Cheng Chen**[1]*, **Hang Wang**[1], **Chen Chen**[1◉], **Ziwei Yan**[1], **Huijie Han**[2], **Fangfang Chen**[1], **Yan Wu**[3], **Zhiao Wang**[1], **Yuxiu Zhou**[1], **Rumeng Si**[1], **Xiaoyi Lv**[1]*

1 College of Information Science and Engineering, Xinjiang University, Urumqi, China, 2 School of Pharmacy, Shanghai Jiao Tong University, Shanghai, China, 3 Quality of Products Supervision and Inspection Institute, Urumqi, Xinjiang, China

◉ These authors contributed equally to this work.
* chenchengoptics@gmail.com (CC); xjuwawj01@163.com (XL)

## Abstract

As a characteristic edible fungus with a high nutritional value and medicinal effect, the *Bachu mushroom* has a broad market. To distinguish among *Bachu mushrooms* with high value and other fungi effectively and accurately, as well as to explore a universal identification method, this study proposed a method to identify *Bachu mushrooms* by Fourier Transform Infrared Spectroscopy (FT-IR) combined with machine learning. In this experiment, two kinds of common edible mushrooms, *Lentinus edodes* and *club fungi*, were selected and classified with *Bachu mushrooms*. Due to the different distribution of nutrients in the caps and stalks, the caps and stalks were studied in this experiment. By comparing the average normalized infrared spectra of the caps and stalks of the three types of fungi, we found differences in their infrared spectra, indicating that the latter can be used to classify and identify the three types of fungi. We also used machine learning to process the spectral data. The overall steps of data processing are as follows: use partial least squares (PLS) to extract spectral features, select the appropriate characteristic number, use different classification algorithms for classification, and finally determine the best algorithm according to the classification results. Among them, the basis of selecting the characteristic number was the cumulative variance interpretation rate. To improve the reliability of the experimental results, this study also used the classification results to verify the feasibility. The classification algorithms used in this study were the support vector machine (SVM), backpropagation neural network (BPNN) and k-nearest neighbors (KNN) algorithm. The results showed that the three algorithms achieved good results in the multivariate classification of the caps and stalks data. Moreover, the cumulative variance explanation rate could be used to select the characteristic number. Finally, by comparing the classification results of the three algorithms, the classification effect of KNN was found to be the best. Additionally, the classification results were as follows: according to the caps data classification, the accuracy was 99.06%; according to the stalks data classification, the accuracy was 99.82%. This study showed that infrared spectroscopy combined with a machine learning algorithm has the potential to be applied to identify *Bachu mushrooms* and the cumulative variance

**Data Availability Statement:** All relevant data are within the manuscript and its Supporting Information files.

**Funding:** This work has been supported by the Science and Technology Project on aid to Xinjiang

Uygur Autonomous Region (No.2018E02058), the National Science Foundation of China (No.61765014), Xinjiang Uygur Autonomous Region Graduate Innovation Project of China, National Innovation Program for College students (No.201910755039), the Urumqi Science and Technology Project (No. P161310002), the Reserve Talents Project of the National Highlevel Personnel of Special Support Program (QN2016YX0324).

**Competing interests:** The authors have declared that no competing interests exist.

explanation rate can be used to select the characteristic number. This method can also be used to identify other types of edible fungi and has a broad application prospect.

## 1. Introduction

The *Bachu mushroom* is a characteristic edible fungus in Xinjiang, China. It belongs to the Saddle fungus genus and is produced in the natural *Populus euphratica* forest region of the Yeerqiang River Basin in Xinjiang [1]. The *Bachu mushroom* not only has high nutritional value, rich in various amino acids and proteins but also has high medicinal value [2]. Studies have shown that the *Bachu mushroom* has anti-tumor, anti-oxidation and cholesterol-lowering effects and is used to treat gastric cancer, cerebral arteriosclerosis, cardiovascular disease, hypertension and other diseases [3]. Its nutritional value is much higher than that of general edible fungi; thus, it is of great research value. However, because the *Bachu mushroom* cannot be cultivated artificially, the market has been in short supply, increasing its price. Presently, the processing technology of the *Bachu mushroom* is developing and relatively mature, such as polysaccharide extraction of the *Bachu mushroom* and preparation of compound beverages, and these processing technologies have a wide application prospect [4–6]. In the future, after the derivatives of the *Bachu mushroom* are mass produced and the treatment process is widely used, controlling the quality of raw materials will be very significant to ensure product quality. Therefore, to prevent businesses from choosing other low-priced mushrooms as raw materials for high profits, it is necessary to identify a simple and rapid way to distinguish *Bachu mushrooms* from other types of mushrooms. However, current methods to identify *Bachu mushrooms* and other edible fungi depend on appearance. This method can distinguish mushroom species to a large extent, but it also has great limitations—that is, it is limited to individual intact edible bacteria—but liquid extract and powdered mushroom powder cannot be distinguished. Thus, to overcome the limitations of conventional methods and explore a more universal mushroom classification method, the spectral data of *Bachu mushroom* powder and two other types of mushroom powder were measured, and the classification of *Bachu mushroom* powder and other types of mushroom powder was identified by a machine learning algorithm in this study.

Due to the differences in the types and contents of nutrients in the stalks and caps of edible fungi, the caps and stalks were analyzed in this study [7]. Previous studies have shown some differences in the contents of proteins and amino acids between caps and stalks in edible fungi [8]. Additionally, the difference in the distribution of nutrients is related to the species of edible fungi [9]. Therefore, to make the experimental results more accurate and persuasive, as well as to avoid the uneven distribution of the substance content in the samples, this study adopted the classification analysis method of grouping according to the attribute index to study the caps and stalks [10].

The research method used in this experiment classified the *Bachu mushroom* and two other types of edible fungi by infrared spectra combined with a machine learning algorithm. Infrared spectra have the characteristics of wide applicability, high efficiency, convenience, repeatability and high sensitivity; thus, it has been widely used in physics, remote sensing, biology, food, medical and other research fields [11, 12]. Infrared spectroscopy has great application value in food research [13, 14]. Additionally, infrared spectroscopy combined with a machine learning algorithm has been applied to the classification of mushroom as food, the effect of the producing area on the nutritional value of fungi and the fine classification of rare edible fungi [15–17]. The purpose of this study was to explore a universal identification method to identify *Bachu mushroom* using infrared spectra combined with a machine learning algorithm and

verify the feasibility of the application of infrared spectra combined with an algorithm in the identification of edible mushroom species.

## 2. Experimental methods

### 2.1. Sample preparation

In this study, *Lentinus edodes* and *club fungi* were selected and classified with *Bachu mushrooms*. Among them, *Lentinus edodes* are produced in Fujian Province of China, and purchased from Fuchang Food Limited Company, Fujian Province of China; *club fungi* are produced in Yunnan Province of China, purchased from Wuweijin Store, and *Bachu mushrooms* are produced in Bachu County, Xinjiang Province, and purchased from the most famous wholesale market in Urumqi—Six Markets. The three kinds of mushrooms purchased are all dried. The caps of the three kinds of mushrooms are umbrella-shaped and dark brown. The stalks of *club fungi* are longer and those of *Lentinus edodes* and *Bachu mushrooms* are shorter. The three kinds of mushrooms are similar in appearance. The samples of the three types of edible fungi were purchased from the market. After incubating the prepared sample in an 80°C electric steam oven for dehydration treatment for 1 hour, the stalks were separated from the caps. Next, every three complete caps were crushed together, and the powder was passed through a 200-mesh sieve as a sample and named according to the mushroom species. The stalks were treated in the same way. Finally, 39 samples of *Lentinus edodes* powder, 47 samples of *club fungi* powder and 35 samples of *Bachu mushroom* powder were obtained.

### 2.2. Measurement of NIF spectra

The sample powder was placed into a 4-ml sample tube, and its infrared spectrum was measured. The spectrum acquisition instrument was a VERTEX 70 infrared spectrometer from BRUKER, Germany. Before each measurement of the FT-IR spectrum, the atmospheric background data were measured using OPUS65 software. The selected resolution was 8 cm$^{-1}$, the number of scans was 32, the scan range was 4000–11000 cm$^{-1}$, and the atmospheric compensation parameter was $CO_2$. To reduce the influence of human error and other factors, each sample was scanned 3 times. Finally, the data obtained for the caps were 117 for *Lentinus edodes*, 141 for *club fungi*, 105 for *Bachu mushrooms*, and the same number of stalks data.

### 2.3. Statistical algorithm analysis

Statistical algorithms have been widely used to manage infrared spectral data [18]. In this study, PLS, SVM, KNN, and BPNN were used to process and analyze the spectral data. The caps data were reduced by PLS to extract features, and then the appropriate characteristic number was selected as the input of the three classification algorithms, namely, SVM, KNN and BPNN, and then the accuracy was obtained. Additionally, the stalk data were processed in the same way. All algorithms in this study are implemented on MATLAB 2018a.

The partial least squares (PLS) method is a mathematical optimization supervised learning method that can identify the best matching function for a set of data by minimizing the sum of the squares of errors. Based on the advantages of the PLS model, which is easy to identify noise and allows regression modeling with a small sample number, the PLS algorithm is widely used in various research fields [7, 19]. In food research, PLS has been used in food nutrition testing, food quality research and food industry research [20–22]. PLS is often used in combination with spectra for feature extraction and further spectral data analysis [23]. In this study, to improve the classification efficiency and filter out the worthless spectral information, PLS was used to reduce the dimensionality of the original spectral data.

After dimensionality reduction of the original data, an appropriate characteristic number should be selected as the basis for classification. In this study, the characteristic number was selected based on the cumulative variance interpretation rate of the characteristic number. The PLS program used in this study is the plsregress function. The variance explanation rates of the factors extracted from the first and second columns of the PCTVAR matrix correspond to the variances of x and y, respectively; this study chose the variance explanation rate of y [24]. The variance interpretation rate is the degree of interpretation of the data characteristics of the dependent variables by a single factor, and the cumulative interpretation rate of n factors is the degree of interpretation of the data characteristics of the dependent variables by n factors— that is, the influence of n factors on the dependent variables. Therefore, in theory, we can select the appropriate number of factors according to the cumulative variance interpretation rate and select as few factors as possible to improve the classification efficiency to ensure the integrity of the extracted features. To explore the applicability of the theory, this study will further use the classification results to verify the theory. After selecting an appropriate characteristic number, it can be used as the input of the classification algorithms SVM, KNN and BPNN.

Support vector machine (SVM) is a commonly used generalized linear classifier, in which the core idea is to apply the principle of risk minimization to the field of classification. Regarding pattern classification, it has good generalization performance, robustness, versatility and simple calculation [25]. Therefore, in food science, SVM is widely applied to food classification and food quality testing [26, 27]. Based on the advantages of SVM and characteristics that can be used for multiple classifications, in this study, SVM was used to classify the spectral data of mushroom powder directly after three features were extracted.

The k-nearest neighbors (KNN) classification algorithm is one of the most practical algorithms in data mining classification technology. It is easy to understand and powerful at the same time [28]. Different from other classification algorithms, KNN does not need training. It directly finds the k samples nearest to the sample and divides them into categories with the largest number of samples among the k samples; thus, KNN is suitable for multivariate classification and has high classification accuracy when the category boundary is obvious [29]. Additionally, KNN has been widely used in food classification and quality inspection [30–32]. Therefore, we chose the KNN algorithm as the second algorithm of multivariate classification.

The backpropagation neural network (BPNN) is a multilayer feedforward neural network trained according to the error backpropagation algorithm. The BP neural network has strong nonlinear mapping ability, parallel information processing ability and excellent self-learning ability; thus, it has been widely used in food research, biomedical fields and other research fields [33–35]. Additionally, the BP neural network can achieve good classification results when it is used in multivariate classification [36]. In summary, we chose the BPNN as the third classification algorithm.

## 3. Results and discussion

### 3.1. Spectral analysis

After the obtained spectral data were averaged, normalized and smoothed, the obtained spectrogram is shown in Fig 1. Fig 1 shows the FT-IR spectra of the *Bachu mushroom* and *Lentinus edodes* stalks were similar, both with characteristic peaks at 5099 cm$^{-1}$ and 8744 cm$^{-1}$, and the spectral intensity of *Lentinus edodes* was higher than that of the *Bachu mushroom*. Additionally, the spectrum of the *Lentinus edodes* stalk has a characteristic peak at 5778 cm$^{-1}$, and the spectral intensity in the range of 8500~11000 cm$^{-1}$ was significantly lower than that of *Lentinus edodes* and the *Bachu mushroom*. Comparing the spectra of the three caps, the spectra of *club fungi* are quite different from those of the other two types of fungi. Fig 2 shows that the average

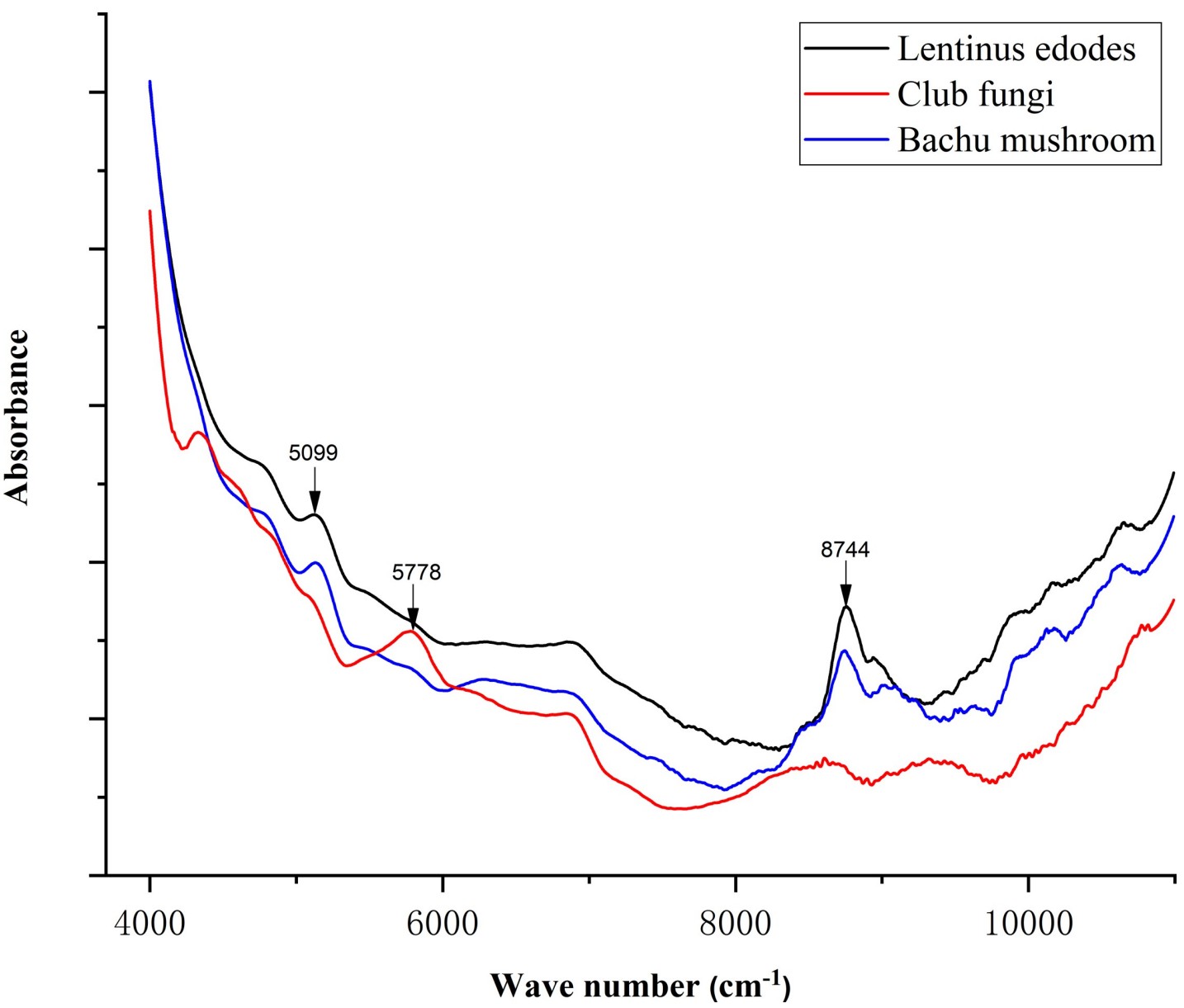

**Fig 1. The average normalized spectrum of the caps of the three types of edible fungi.**

normalized spectra of *Lentinus edodes*, *club fungi* and *Bachu mushroom* stalks all had peaks at 5153 cm$^{-1}$ and 8755 cm$^{-1}$, and the spectral intensity of *Lentinus edodes* was the highest in these two places. At 5153 cm$^{-1}$, the peak intensity of *club fungi* was higher than that of the *Bachu mushroom*; however, at 8755 cm$^{-1}$, the peak intensity of *club fungi* was slightly lower than that of the *Bachu mushroom*. Although the three spectral lines changed roughly the same; however, in the range of 4300~6800 cm$^{-1}$, the spectral intensity of the *Bachu mushroom* was significantly lower than that of the other two kinds of fungi, and the spectral intensity of the *Bachu mushroom* was significantly lower than that of the other two kinds of fungi.

Through comparative analysis of the infrared spectra of the caps and stalks of the three types of fungi, their infrared spectral images showed the same trend, but many peak intensities were different. Therefore, we can classify them according to the spectral data based on the

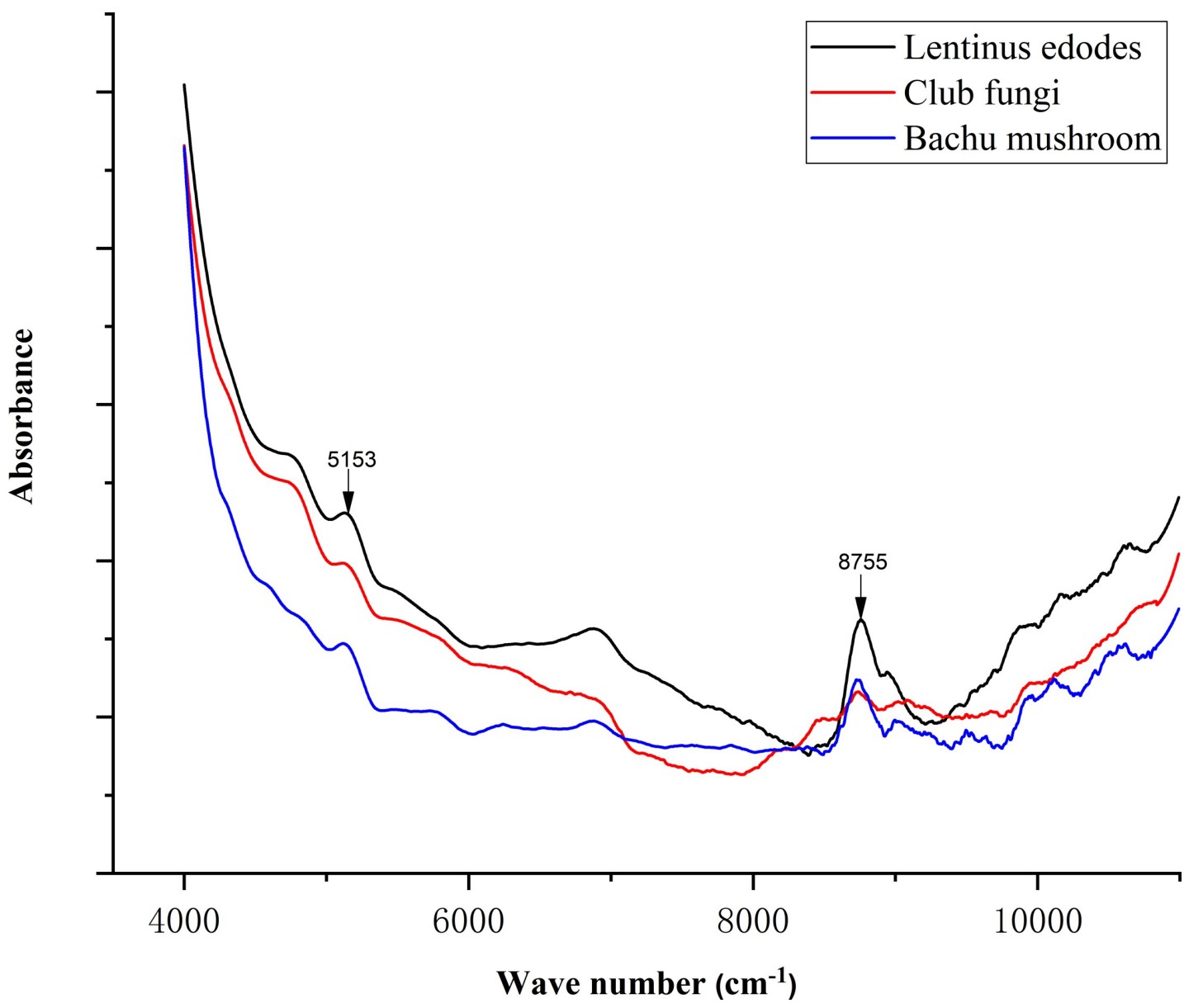

**Fig 2. The average normalized spectrum of the stalks of the three types of edible fungi.**

infrared spectral differences among the three types. However, it is difficult to distinguish among the three types of edible fungi directly and accurately only by spectroscopy. Thus, to classify them efficiently and accurately, we used the combined infrared spectrum analysis with machine learning.

## 3.2. Data analysis

**3.2.1. Dimensionality reduction with PLS.** In the PLS algorithm, 50 features were selected to obtain the cumulative variance explanation rate curve (Figs 3 and 4). The cumulative variance explanation rate of the first five features of the caps reached 90%, while the variance explanation rate of the first five features of the stalks also reached more than 80%, and the

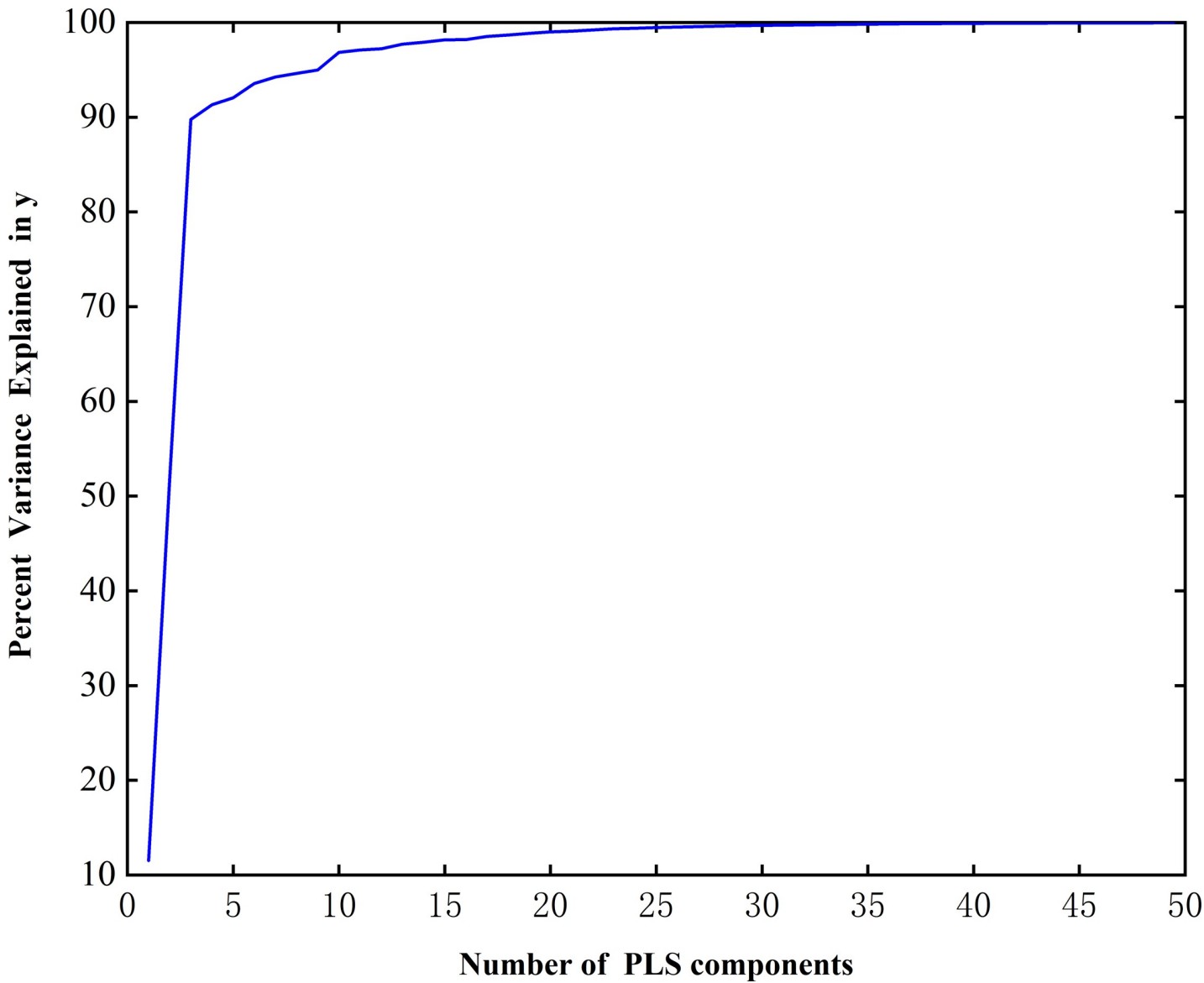

**Fig 3. The principal component cumulative variance explanation rate curve of the caps.**

cumulative variance explanation rate of the first 30 features of both was close to 100% (Figs 3 and 4). Thus, the extracted features can fully express the features of the original data [37]. The stalks data and caps data were classified by the SVM, BPNN, and KNN algorithms.

**3.2.2. Classification by algorithm.** In this experiment, three classification algorithms were used—SVM, BPNN, and KNN—to classify the caps data according to different characteristic numbers. Additionally, the stalk data were processed in the same way. The parameter setting and classification results of the algorithm were as follows:

The main ideas of the SVM model in this experiment were as follows: Select the test set and training set; preprocess the data; select the best C and g parameters, and then use the best parameters for network training and prediction; and obtain the accuracy. Among them, the training set and test set are randomly selected according to the proportion 7:3. Pre-processing was used to normalize all the sample data [0,1]. In the SVM algorithm, the selection of C and g

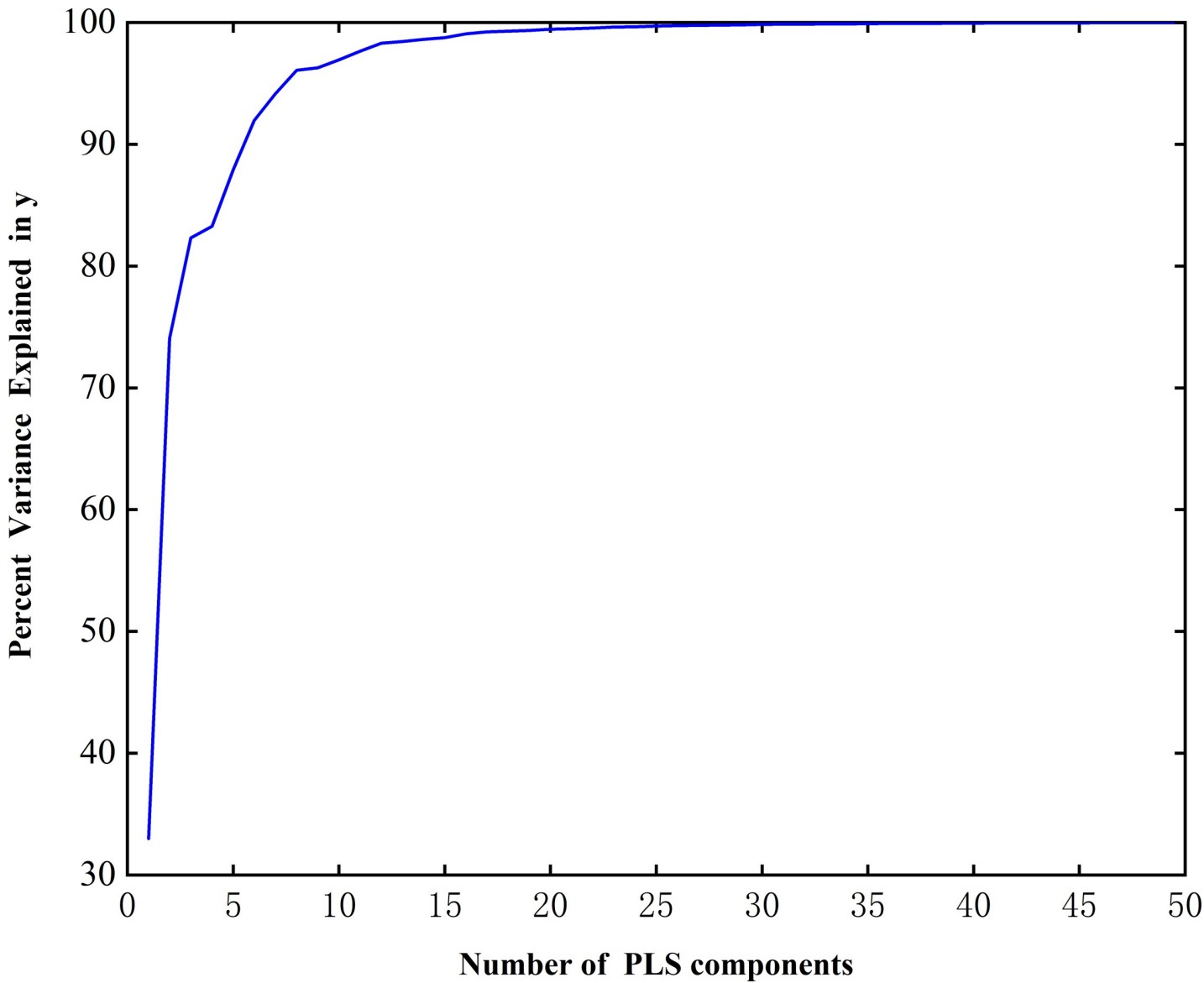

**Fig 4. The principal component cumulative variance explanation rate curve of the stalks.**

parameters will directly affect the classification results; thus, it is necessary to select the best C and g parameters to achieve the best classification results. In this study, the variation range of parameter C was $[2^{-2},2^4]$, the range of parameter g was $[2^{-4},2^4]$, and the method of parameter optimization was grid optimization [38]. To classify the caps data, 5, 10, 15, 20, 25, 30, 35, 40, 45 and 50 features were selected. The stalks data were processed in the same way. The multiple classification results of the stalks and caps were then obtained, as shown in Table 1.

In the KNN algorithm, the k value was 5, the proportion of the random selection of the test set was 30%, and the method to calculate the distance between data was the cosine distance (Cosine KNN) [39]. To classify the caps data, 5, 10, 15, 20, 25, 30, 35, 40, 45 and 50 features were selected. The stalks data were processed in the same way. Each result was expressed as the average of five computational results. The multiple classification results of the stalks and caps were then obtained, as shown in Table 2.

**Table 1. Classification results of the SVM algorithm.**

| Number of PLS components | Classification accuracy of Caps | Classification accuracy of Stalks |
|---|---|---|
| 5 | 100.00% | 79.63% |
| 10 | 100.00% | 81.48% |
| 15 | 100.00% | 80.56% |
| 20 | 98.15% | 82.41% |
| 25 | 99.07% | 82.41% |
| 30 | 98.15% | 83.33% |
| 35 | 98.15% | 83.33% |
| 40 | 98.15% | 83.33% |
| 45 | 98.15% | 81.48% |
| 50 | 98.15% | 81.48% |

In the BPNN algorithm of this experiment, the transfer function of the hidden layer was tamsig, the output layer was purelin, the learning training function was trainlm, and the weight learning function was learngdm. The network parameters were set to 300 training times, the network performance goal was 0.1, and the learning rate was 0.1 [40]. Thirty percent of all samples were selected randomly as the test set. To classify the caps data, 5, 10, 15, 20, 25, 30, 35, 40, 45 and 50 features were selected. The stalks data were processed in the same way. Each result was expressed as the average of five computational results. The multiple classification results of the stalks and caps were then obtained, as shown in Table 3.

**3.2.3. Verification of the feasibility of selecting the characteristic number with the cumulative variance explanation rate.** Figs 5 and 6 shows the line chart of the classification results of the three algorithms when selecting different characteristic numbers. The accuracy of SVM in the classification of caps was 100% when selecting 5, 10 and 15 features, decreased slightly with the increase in the characteristic number, and then stabilized at 98%. According to the fungal stalks data, the accuracy of SVM classification increased gradually when selecting 5, 10, 15, 20 features, fluctuated slightly with the increase in the characteristic number, and then stabilized at 82%. Using KNN to classify the caps data, the accuracy was stable between 97.40% and 99.07% but fluctuated slightly. When the characteristic number tended to 50, the accuracy was stable at 97.40%. According to the classification of stalks data by KNN, the accuracy varied greatly when selecting 5–10 features, fluctuated slightly with the increase in the characteristic number, and then stabilized at 99.50%. For the classification of caps by BPNN, the accuracy varied greatly when the characteristic number was 5~30, and the classification

**Table 2. Classification results of the KNN algorithm.**

| Number of PLS components | Classification accuracy of Caps | Classification accuracy of Stalks |
|---|---|---|
| 5 | 98.14% | 95.36% |
| 10 | 98.89% | 100.00% |
| 15 | 99.06% | 99.82% |
| 20 | 98.12% | 99.82% |
| 25 | 98.89% | 99.82% |
| 30 | 98.90% | 100.00% |
| 35 | 98.14% | 100.00% |
| 40 | 98.32% | 99.82% |
| 45 | 97.40% | 99.46% |
| 50 | 97.40% | 99.44% |

**Table 3. Classification results of the BPNN algorithm.**

| Number of PLS components | Classification accuracy of Caps | Classification accuracy of Stalks |
|---|---|---|
| 5 | 94.26% | 93.33% |
| 10 | 92.24% | 94.81% |
| 15 | 92.96% | 94.08% |
| 20 | 95.56% | 98.70% |
| 25 | 99.07% | 94.07% |
| 30 | 98.15% | 95.56% |
| 35 | 97.96% | 93.15% |
| 40 | 97.78% | 93.70% |
| 45 | 97.59% | 93.70% |
| 5 0 | 97.59% | 93.52% |

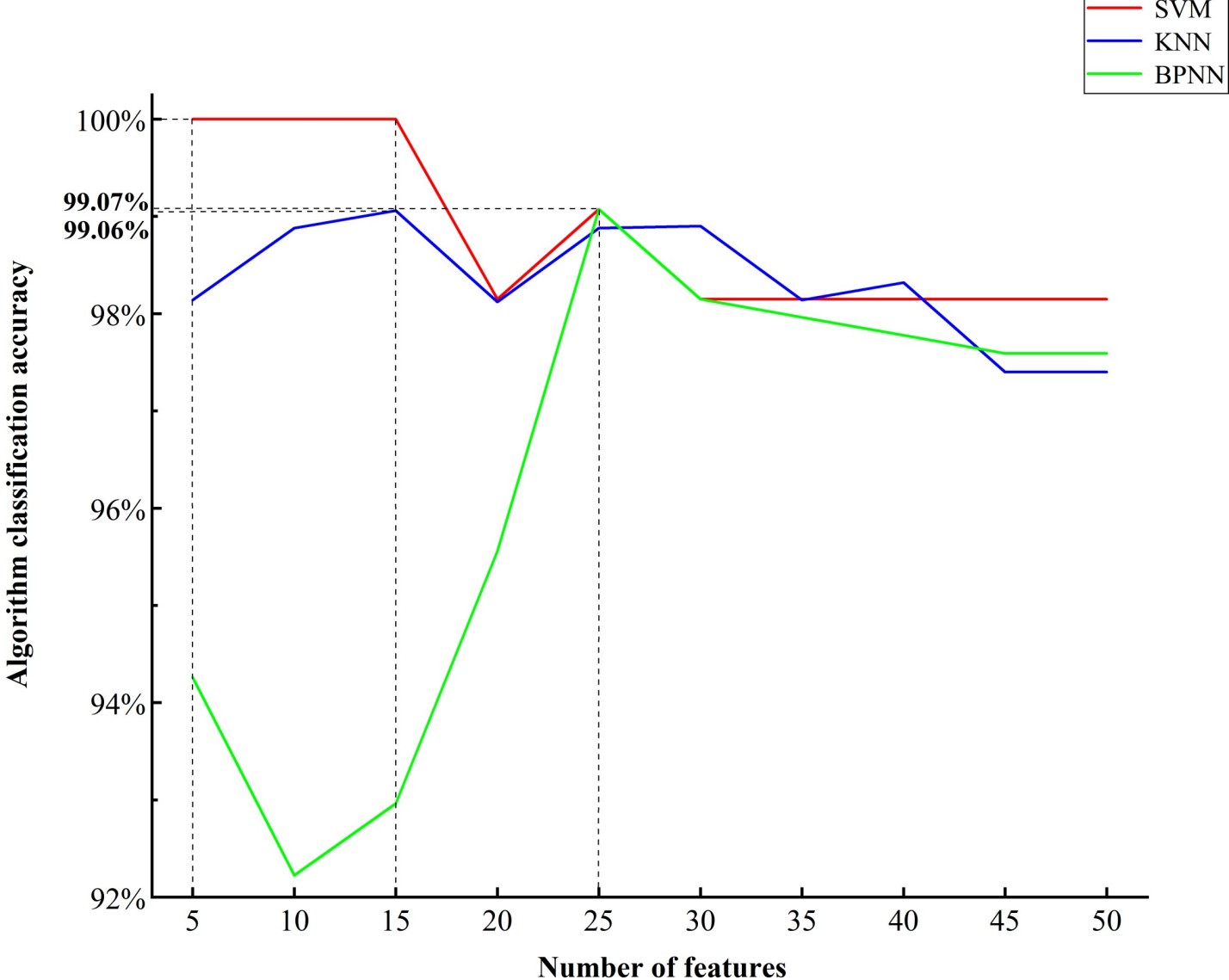

**Fig 5. Accuracy of the three algorithms to select different characteristic numbers according to the data of the caps.**

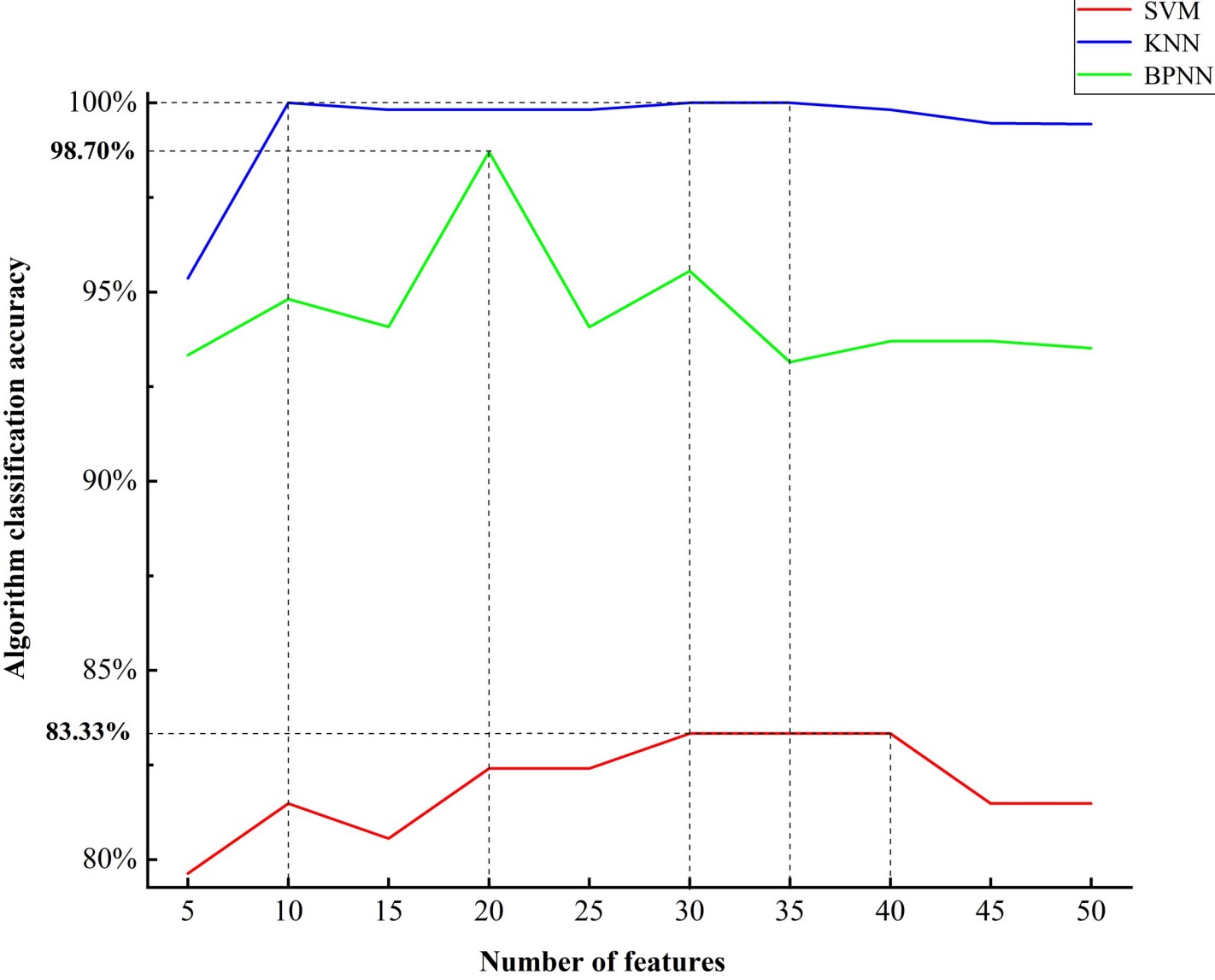

**Fig 6. Accuracy of the three algorithms to select different characteristic numbers according to the data of the stalks.**

effect was unstable. When the characteristic number was more than 30, the accuracy decreased slightly and finally stabilized at 97.59%. When BPNN classified stalks, its accuracy varied greatly when the characteristic number was less than 35; when the characteristic number was more than 35, the accuracy was stabilized at 93.6%.

Combined with the cumulative variance explanation rate, with the increase in the characteristic number, the extraction degree of the extracted features to the original data information gradually increased. Thus, the classification results become more reliable, and the accuracy will gradually tend to a certain value—that is, in the ideal state, the accuracy of all features extracted for classification. However, in this process, not all information is conducive to improving accuracy and some information will interfere with the classification results; thus, the accuracy will fluctuate slightly [41]. Therefore, when we select the characteristic number according to the variance interpretation rate, we should consider the degree of feature

extraction, accuracy and classification efficiency comprehensively. In summary, when the cumulative variance interpretation rate of features reaches 90%, the classification results are sufficiently reliable and the accuracy is high, confirming that it is feasible to select the characteristic number according to the cumulative variance interpretation rate.

**3.2.4. Selection of the best algorithm.** According to the caps data classification, the PLS-SVM algorithm has the best classification effect when selecting 5~15 features, with an accuracy of 100%. The PLS-KNN algorithm has the best classification effect when selecting 15 features, with an accuracy of 99.06%. When selecting 25 features, the PLS-BPNN algorithm has the best classification effect, with an accuracy of 99.07% (Fig 5). According to the stalks data classification, the classification effect of the PLS-SVM algorithm was the best when selecting 30~40 features, with an accuracy of 83.33%. The classification effect of the PLS-KNN algorithm was the best when selecting 35 features, with an accuracy of 100%. When 20 features were selected, the classification effect of the PLS-BPNN algorithm was the best, reaching 98.70% (Fig 6).

Combined with the selection of the characteristic number to analyze the three algorithms, the PLS-KNN algorithm has a better classification effect on stalks and caps, and the accuracy is more stable when selecting different characteristic numbers. Thus, the PLS-KNN algorithm was chosen as the optimal algorithm. Using comprehensive analysis of the accuracy of PLS-KNN classification when selecting different characteristic numbers, a characteristic number of 15 reveals a higher accuracy for both caps and stalks and a high classification efficiency. Therefore, in this experiment, the PLS-KNN algorithm was finally selected, and the characteristic number was 15. The final classification accuracy was 99.06% for the caps and 99.82% for the stalks.

## 4. Conclusions

This study verified the feasibility of infrared spectroscopy combined with the PLS-SVM, PLS-KNN and PLS-BPNN algorithms in the classification of the *Bachu mushroom* and other edible mushrooms. We compared the classification results and selected the optimal algorithm and best feature number to reveal an efficient, rapid and universal method to identify the *Bachu mushroom*, overcoming the limitation that the current identification of *Bachu mushroom* only depends on its appearance. Moreover, the method is universal and can be applied to the classification and identification of other types of food. Additionally, this study proposed to select the characteristic number according to the cumulative variance interpretation rate and used the classification results of the three algorithms to verify its feasibility. This method of selecting the characteristic number can also be extended to other research fields of factor analysis, and the appropriate characteristic number can be selected intuitively and quickly.

## Supporting information

**S1 Data.**
(ZIP)

## Author Contributions

**Conceptualization:** Cheng Chen, Zhiao Wang.

**Data curation:** Rui Gao, Cheng Chen, Hang Wang, Chen Chen, Ziwei Yan, Yan Wu.

**Formal analysis:** Hang Wang.

**Funding acquisition:** Cheng Chen, Chen Chen, Fangfang Chen, Zhiao Wang.

**Investigation:** Rui Gao, Fangfang Chen.

**Methodology:** Rui Gao, Cheng Chen, Chen Chen, Yuxiu Zhou.

**Project administration:** Rui Gao, Cheng Chen, Rumeng Si, Xiaoyi Lv.

**Resources:** Rui Gao, Ziwei Yan, Fangfang Chen, Yan Wu.

**Software:** Ziwei Yan.

**Supervision:** Cheng Chen.

**Validation:** Rui Gao, Rumeng Si.

**Writing – original draft:** Rui Gao, Hang Wang, Zhiao Wang, Yuxiu Zhou.

**Writing – review & editing:** Rui Gao, Cheng Chen, Huijie Han, Xiaoyi Lv.

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
