## [Decision Letter · Decision Letter 0]

8 Jul 2020

PONE-D-20-12067

Classification of Multicategory Edible Fungi based on the Infrared Spectra of Caps and Stalks

PLOS ONE

Dear Dr. Lv,

Thank you for submitting your manuscript to PLOS ONE. After careful consideration, we feel that it has merit but does not fully meet PLOS ONE’s publication criteria as it currently stands. Therefore, we invite you to submit a revised version of the manuscript that addresses the points raised during the review process.

Specifically, please specify which version of Matlab was used and to italicise the biological genus and species names of the fungi.

We look forward to receiving your revised manuscript.

Kind regards,

Jie Zhang

Academic Editor

PLOS ONE

Journal Requirements:

1. We understand that you purchased mushrooms from local markets for this study. In your Methods section, please provide additional details regarding the source of this material.

Please provide the geographic coordinates and names of the purchase locations (e.g., stores, markets), if available, as well as any further details about the purchased items (e.g., lot number, source origin, description of appearance) to ensure reproducibility of the analyses.

Additional Editor Comments (if provided):

Reviewers' comments:

Reviewer's Responses to Questions

**Comments to the Author**

1. Is the manuscript technically sound, and do the data support the conclusions?

Reviewer #1: Yes

2. Has the statistical analysis been performed appropriately and rigorously? 

Reviewer #1: Yes

3. Have the authors made all data underlying the findings in their manuscript fully available?

Reviewer #1: Yes

4. Is the manuscript presented in an intelligible fashion and written in standard English?

Reviewer #1: Yes

5. Review Comments to the Author

Reviewer #1: The manuscript is very well written all the statistical analysis have been done with great rigour and very nicely presented. The manuscript is very easy to follow and targeted for chemometricians. My only comments is to specify which version of Matlab was used and to italicise the biological genus and species names of the fungi. Good job once again to all the authors involved in this publication.

6. PLOS authors have the option to publish the peer review history of their article (what does this mean?). If published, this will include your full peer review and any attached files.

Reviewer #1: **Yes: **Frederick Lia

---

## [Author Response · Author response to Decision Letter 0]

10 Aug 2020

Subject: Revised version of the manuscript

No.: PONE-D-20-12067

Title: Classification of Multicategory Edible Fungi based on the Infrared Spectra of Caps and Stalks

Dear Editor and Reviewer:

Thanks to the editor and reviewer for your patient review and important suggestion for our article. We have further revised your valuable suggestions. We have highlighted our manuscript changes through the use of red font in the revised manuscript. 

The revision was carried out as a result of the comments of the editor and the reviewer.

The main corrections in the paper and the responds to the editor’s comments and the responds to the reviewer’s comments are as following:

Editor :

Response to comment (1):

( We understand that you purchased mushrooms from local markets for this study. In your Methods section, please provide additional details regarding the source of this material. Please provide the geographic coordinates and names of the purchase locations (e.g., stores, markets), if available, as well as any further details about the purchased items (e.g., lot number, source origin, description of appearance) to ensure reproducibility of the analyses.)

Response-1: Many thanks to the editor for the suggestions. Based on your suggestions, we have supplemented the description of the source origin, names of the purchase locations and appearance of mushrooms in 2. Experimental methods to ensure reproducibility of the analyses. Thanks again to the editor for your valuable suggestions. 

The contents are as follows: 

Among them, Lentinus edodes are produced in Fujian Province of China, and purchased from Fuchang Food Limited Company, Fujian Province of China; club fungi are produced in Yunnan Province of China, purchased from Wuweijin Store, and Bachu mushrooms are produced in Bachu County, Xinjiang Province, and purchased from the most famous wholesale market in Urumqi -- Six Markets. The three kinds of mushrooms purchased are all dried. The caps of the three kinds of mushrooms are umbrella-shaped and dark brown. The stalks of club fungi are longer and those of Lentinus edodes and Bachu mushrooms are shorter. The three kinds of mushrooms are similar in appearance. The samples of the three types of edible fungi were purchased from the market.

Reviewer #1:

1.Response to comment (1-4):

Response: Thank you very much for the reviewer's agreement on our work, which will encourage us to do better work in the future.

2.Response to comment (5):

( The manuscript is very well written all the statistical analysis have been done with great rigour and very nicely presented. The manuscript is very easy to follow and targeted for chemometricians. My only comments is to specify which version of Matlab was used and to italicise the biological genus and species names of the fungi. Good job once again to all the authors involved in this publication.)

Response-5: Thanks to the reviewer for the suggestions. We have supplemented the MATLAB version and italicised the biological genus and species names of the fungi in the article.

Thank the reviewer and editor for your patient and careful review. I hope you have good health and success in your work.

Sincerely yours,

Dr. Xiaoyi Lv, Dr. Cheng Chen

---

## [Editor Report · Decision Letter 1]

11 Aug 2020

Classification of Multicategory Edible Fungi based on the Infrared Spectra of Caps and Stalks

PONE-D-20-12067R1

Dear Dr. Lv,

We’re pleased to inform you that your manuscript has been judged scientifically suitable for publication and will be formally accepted for publication once it meets all outstanding technical requirements.

Kind regards,

Jie Zhang

Academic Editor

PLOS ONE
---

## [Editor Report · Acceptance letter]

13 Aug 2020

PONE-D-20-12067R1 

Classification of Multicategory Edible Fungi based on the Infrared Spectra of Caps and Stalks 

Dear Dr. Lv:

I'm pleased to inform you that your manuscript has been deemed suitable for publication in PLOS ONE. Congratulations! Your manuscript is now with our production department. 

Kind regards, 

on behalf of

Dr. Jie Zhang 

Academic Editor

PLOS ONE